# Magnetic Anisotropy and Microstructure in Electrodeposited Quaternary Sn-Fe-Ni-Co Alloys with Amorphous Character

**DOI:** 10.3390/ma15093015

**Published:** 2022-04-21

**Authors:** Ernő Kuzmann, Israel Felner, Laura Sziráki, Sándor Stichleutner, Zoltán Homonnay, Mahmoud R. El-Sharif, Colin U. Chisholm

**Affiliations:** 1Institute of Chemistry, Eötvös Loránd University, Pázmány P. s. 1/A, 1117 Budapest, Hungary; sziraki.laura@ttk.elte.hu (L.S.); stichleutner.sandor@ek-cer.hu (S.S.); homonnay.zoltan@ttk.elte.hu (Z.H.); 2Racah Institute of Physics, The Hebrew University, Jerusalem 91904, Israel; israel.felner@mail.huji.ac.il; 3Centre for Energy Research, Konkoly-Thege Miklós út 29–33, 1121 Budapest, Hungary; 4Glasgow Caledonian University, Cowcaddens Road, Glasgow G4 0BA, UK; melsharifraj@gmail.com (M.R.E.-S.); colinchisholm357@gmail.com (C.U.C.)

**Keywords:** Sn-Fe-Ni-Co quaternary alloys, magnetic anisotropy, saturation magnetization, amorphous alloys, ^57^Fe and ^119^Sn conversion electron Mössbauer spectroscopy, electrodeposition

## Abstract

Sn-Fe-Ni-Co quaternary alloys, in the composition range of 37–44 at% Sn, 35–39 at% Fe, 6–8 at% Ni and 13–17 at% Co, were prepared by direct current (DC) and pulse plating (PP) electrodeposition. The alloy deposits were characterized by XRD, ^57^Fe and ^119^Sn conversion electron Mössbauer spectroscopy, SEM-EDX and magnetization measurements. XRD revealed the amorphous character of the quaternary alloy deposits. The dominant ferromagnetic character of the deposits was shown by magnetization and Mössbauer spectroscopy measurements. Room temperature Mössbauer spectra showed minor paramagnetic phases, where their occurrences (~3–20%) are correlated to the electrodeposition parameters (*J*_dep_ from −16 to −23 mA/cm^2^ for DC, *J*_pulse_ from −40 to −75 mA/cm^2^ for PP), the composition and the saturation magnetization (~52–73 emu/g). A considerable difference was found in the magnetization curves applying parallel or perpendicular orientation of the applied fields, indicating magnetic anisotropy both in DC and pulse plated alloy coatings.

## 1. Introduction

The application of electrodeposition to produce compositionally modulated materials is attracting considerable interest at present because of all the processing techniques that are available, electrodeposition is the simplest and the most economical process. Moreover, it has the minimum number of technological barriers in transferring the technique from the laboratory to the existing electroplating and electroforming industries.

In our earlier works [1,2,3,4,5,6,7,8,9], systematic studies were conducted to understand the effect of preparation parameters on the electrodeposited alloy Sn-Fe [1,3], Sn-Co-Fe [2,3,4], Sn-Ni-Fe [5,6,7] and Sn-Fe-Co-Ni [8,9] systems, on the structure, the phase composition and the chemical short-range order of the coatings, in order to find the optimal parameters for co-deposition of metals which form non-equilibrium phases. In most cases, metastable amorphous phases were found in the electrodeposits, while for the thermally prepared alloys, these phases do not appear in the associated phase diagrams. Electrodeposited Sn-based alloy coatings were shown to be promising as an alternative anode material for lithium ion batteries [10,11].

Mössbauer spectroscopy proved to be a useful technique to characterize the phase composition, chemical short-range order, magnetic anisotropy and the formation of metastable phases in electrochemically deposited alloys [1,2,3,4,5,6,7,8,9]. ^57^Fe Mössbauer spectra of all the studied Sn-containing ternary and quaternary alloy coatings exhibited dominantly magnetically split spectra with broad lines, typical of the ferromagnetic and amorphous character of the deposits. Magnetically split ^119^Sn Mössbauer spectra of these amorphous coatings were also observed due to the transferred hyperfine fields, being consistent with their ferromagnetic nature. A monotonous change in the magnetic anisotropy was also observed with the current densities used in the electrodeposition studies. The analysis of Mössbauer spectra of the deposits of different composition indicated that the short-range order in direct current (DC) deposited ferromagnetic amorphous binary Sn-Fe, ternary Sn-Co-Fe, Sn-Ni-Fe and quaternary Sn-Fe-Co-Ni alloys can be considered as nearly random distribution of the constituting elements. The short-range order of pulse plated amorphous quaternary Sn-Fe-Co-Ni alloys deviated from that of the DC deposited alloys.

In order to obtain further information on the structure and magnetic properties of these deposits, it seemed expedient to supplement our studies with magnetic measurements.

The study of the magnetic properties of Sn-Fe alloys was reported 50–60 years ago [12,13,14]. Amorphous Fe_x_Sn_1−x_ alloys were found to be ferromagnets with a mixture of magnetic and nonmagnetic iron atoms below a critical composition [15,16]. Concentration-dependent changes in magnetic behavior in amorphous Co_x_Sn_1−x_ alloys were associated with changes in the short-range order [17]. Based on bulk magnetic data obtained in Fe_x_Sn_1−x_ amorphous alloys around the critical composition, a multicritical magnetic phase diagram with paramagnetic, ferromagnetic and spin-glass-like phases was proposed [18]. The coexistence of ferromagnetic and spin-glass structures was suggested in mechanically grinded nano-crystalline Fe_1−x_Sn_x_ alloys [19]. In polycrystalline Fe-Sn alloys, the observed magnetostriction was attributed to the presence of Fe_5_Sn_3_ or Fe_3_Sn_2_ phases [20]. In crystalline ferromagnetic Fe_3_Sn and related alloys, magnetic anisotropy was observed via the difference between the curves of magnetization versus applied field when the field was applied parallel or perpendicular [21] where the easy axis of magnetization was found to be in the hexagonal plane. In crystalline Fe_3_Sn_2_, a uniaxial anisotropy parallel to the c axis was observed, which rotated into the hexagonal plane at low temperatures [22,23]. An increase in magnetic anisotropy was shown when high magnetic fields were applied in the solidification process of Fe–49%Sn monotectic alloys [24]. A significant difference was found between their hysteresis loops measured parallel and perpendicular to the applied high magnetic fields. The saturation magnetization increased along the parallel directions, which was attributed to the aligned microstructure of Fe–Sn alloys [24].

In crystalline YbMn_6−y_Fe_y_Sn_6_ alloys (y = 0.25, 0.50, 0.75 and 1.00), it was established that the compound with y = 0.25 is a pure ferromagnet over the whole ordered temperature range, while the alloys richer in Fe evolve upon cooling towards a low-magnetization antiferromagnetic state through a first-order transition [25]. It was observed that the CrMnFeCoNi high-entropy alloy undergoes two magnetic transformations at temperatures below 100 K while maintaining its face-centered-cubic (FCC) structure down to 3 K. The first transition, paramagnetic to spin glass, was detected at 93 K and the second transition to the ferromagnetic type occurred at 38 K [26]. Hysteresis loops for Ni_2_FeSn ribbons, prepared by rapid quenching, revealed ferromagnetic and anisotropic character, where the easy magnetization plane is parallel to the ribbon surface [27].

The main objective of our present work is to determine the magnetic properties of the Sn-Fe-Ni-Co quaternary alloy coatings, produced by the DC and pulse plating (PP) techniques, as functions of the preparation parameters. For this, we performed magnetization measurements at various temperatures and magnetic fields parallel and perpendicular to coating surfaces, X-ray diffractometry (XRD), scanning electron microscopy and energy-dispersive X-ray spectroscopy (SEM-EDX), ^57^Fe and ^119^Sn conversion electron Mössbauer spectroscopy (CEMS) measurements.

## 2. Experimental Procedure

Samples of quaternary alloy (Sn-Fe-Ni-Co) coatings were prepared by electrochemical deposition onto freshly electropolished 0.2 mm thick Cu substrate plates with a total surface of 4–5 cm^2^ by a similar way as described in previous works [5,6,9]. The metallic coatings of a few micrometers in thickness covered both sides of Cu substrate sheets. In order to achieve the co-deposition of Sn, Fe, Co and Ni together, i.e., to obtain quaternary Sn-Fe-Ni-Co alloy coatings, the electrodeposition was conducted using a dilute plating bath containing a complexing agent, a high overvoltage and a high current density at moderate pH, similar to previous work [5,6,7,8,9]. The electrolyte bath composition used for all the deposited alloys is shown in Table 1 and the selected plating parameters are depicted in Table 2, which also presents the chemical composition of the samples determined by EDX analysis and the thickness of the deposits, which was calculated using the plating parameters.

The electrodeposition was performed with either galvanostatic (DC) or pulse current (PP) modes in a two-compartment cell using the AutoLab PGSTAT 302N electrochemical work station with Nova 2.1.2software (Metrohm Autolab, Utrecht, The Netherlands). The details of the set-up of the plating cell and electrode pretreatment methods were reported previously [5,6,8]. Saturated calomel electrode (SCE) (0.24 V compared to hydrogen electrode) was used as a reference electrode equipped with a Luggin capillary placed close to the cathode surface. The Luggin capillary was equipped with a glass frit in the tip filled with 1 mol/dm^3^ Na_2_SO_4_ solution. Two high-density carbon sheets were placed in parallel at 1–1 cm distance from the cathode. The electrolytes were circulated using a strong magnetic stirrer. Electrodeposition studies were performed at 20 °C using an electrolyte with a pH of 5.1. In the DC plating mode, 15 min plating times were applied. Both DC and PP deposition occurred together with hydrogen evolution.

The galvanostatic square pulses were applied with varying pulse times and duty cycles at the same time period of 0.2 s. The plating was performed at pulse-on times of 0.01–0.02–0.03 s and pulse-off times of 0.19–0.18–0.17 s for a pulse frequency of 5 Hz. The plating duration was 600 s.

X-ray diffractograms of the electrodeposits were recorded with a DRON-2 computer controlled diffractometer (Saint Petersburg, Russia) at room temperature in Bragg–Brentano geometry. The XRD patterns were measured with β-filtered Fe_Kα_ radiation (λ = 0.193735 nm) (at 45 kV and 35 mA) in the range of 2θ = 20–70 degrees with a goniometer speed of 0.25 degree/min. EXRAY peak searching software (developed by Z. Klencsár, 1996) was used for the evaluation of the X-ray diffractograms. For identification of the phases, the PCPDF Diffraction Data (International Centre for Diffraction Data, Newtown Square, PA, USA) were used.

The FEI Quanta 3D (Hillsboro, OR, USA) high-resolution scanning electron microscope was used for the SEM measurements and elemental composition was measured by the EDAX Apollo XP SDD EDS (EDAX, Mowo, NJ, USA) detector.

^57^Fe and ^119^Sn conversion electron Mössbauer spectra (CEMS) of the electrodeposits were recorded with conventional Mössbauer spectrometers (WISSEL type, Wissel-GmbH, Starnberg, Germany). The measurements were made in constant acceleration mode using an integrated multichannel analyzer, in reflection geometry. The CEMS spectra were measured at room temperature by a flowing gas RANGER type detector (Ranger Electronic Corporation, Pittsburg, PA, USA) using He-4% CH_4_ gas mixture. A ^57^Co source of 0.8 GBq activity in Rh matrix supplied the gamma rays for ^57^Fe measurements, while a 0.1 GBq activity Ca^119m^SnO_3_ source was applied for the ^119^Sn measurements. α-Fe measurement was used for the velocity calibration. The isomer shifts are given relative to α-Fe and to CaSnO_3_ for ^57^Fe and ^119^Sn spectra, respectively. The evaluations of the Mössbauer spectra were performed by least-square fitting of individual spectral lines and by deriving hyperfine field distributions (Hesse-Rübartsch method) by the MOSSWINN software [28].

Magnetization measurements of electrodeposits were performed at the Hebrew University, using a commercial SQUID magnetometer MPMS-5S (Quantum Design, Darmstadt, Germany) in the temperature range between 5 and 300 K. The isothermal field-dependent magnetic moment *m*(*H*) curves were measured at 5 and 295 K with applied magnetic fields (*H*) up to 32 kOe. During the measurements, the surface of the deposits was oriented parallel and perpendicular to the applied external magnetic field, *H*. The *m*(*H*) values were normalized to the electrodeposits areas and exhibited as emu/*d*, where *d* is the thickness in μm, as listed in Table 2.

## 3. Results and Discussion

### 3.1. Electrochemical and Morphological Characterization

In Figure 1, the cathode potential curve versus the deposition time for the DC deposits shows only a small change in deposition potentials over time, indicating a stable bath operation and the fulfillment of a stationary plating condition.

The small potential time dependence of the partial deposition currents suggested that the alloy components are deposited under diffusion control, which is influenced by the potential dependence of the co-deposition rate on the charge transfer controlled hydrogen evolution, similarly to that reported in our previous work [9].

In the case of the PP deposition, the effect of decreasing pulse time on the thickness of the pulsating diffusion layer causes a significant increase in the diffusion current. The duty cycle affects the average plating current for the metals, and this characterizes the magnitude of the diffusion rate in the stationary outer diffusion layer [29].

In the potential decays shown in Figure 2, a sudden decrease in potential due to the double layer discharge process, then the slow relaxation of the potential to the rest potentials are typical for the total deposition time. Figure 2 shows that the potential relaxation is slower with increasing pulse-on time, *t*_on_. The small difference in the rest potentials at the end of the period indicates that the recovery of the electrode surface and the concentration of the metals can vary with these pulse times. By calculating using the data from Table 2, it was found that the PP plating rates during the applied pulse times are about 1 order of magnitude larger than those of the DC samples. The differences in the potential decays during *t*_off_ time show that the state of the electrode surface changes periodically and is different from the results obtained with the DC deposition results. It was expected that the structure of the deposits would be different for the DC and PP depositions.

The elemental composition of the samples obtained from the SEM-EDX analysis are given in Table 2, which reflects similar dependences on deposition parameters, as reported earlier [9].

Typical SEM micrographs of the DC and PP deposits (see Table 2) are depicted in Figure 3 and Figure 4, respectively. By comparing the micrographs, it can be seen that there is a large difference between the morphology of the DC and PP deposits. While deposits obtained using DC deposition show a homogeneous, uniform, structureless surface with microcracks, the morphological structure of the pulse plated coatings are characterized by spherical structures of varying size. The morphological differences may indicate differences in the short-range ordering of DC and PP deposits.

### 3.2. X-Ray Diffraction Study of the Coating’s Structure

Figure 5 shows the details of X-ray diffractograms of DC and PP quaternary alloys electrodeposited on copper substrate.

All diffractograms consist of very broad lines centered mainly around 2θ = 40° and sharp lines. The broad lines are associated with the amorphous metallic phase. The sharp lines belonging to the reflections of Cu substrate and the (111)_Kα_ and (200)_Kβ_ Cu reflections at 2θ = 55.3° and 2θ = 58.1°, respectively, can be seen in Figure 5. The occurrence of Cu reflections in X-ray diffractograms is due to the fact that the thickness of the deposits (~1–5 μm, see Table 2) is much smaller than the X-ray penetration depth (~20 μm); thus, the pattern of Cu substrate is strongly reflected.

The diffractograms shown in Figure 5 are similar to those obtained for the Sn-containing amorphous coatings deposited on a Cu substrate [5,6,7,8,9]. Consequently, the X-ray diffractograms of all deposits presumably show that amorphous structures could be successfully achieved in both the DC and PP plated cases. This demonstrates that the applied deposition parameters were suitable for the preparation of quaternary Sn-Fe-Ni-Co alloy coatings.

### 3.3. Mössbauer Study of the Quaternary Sn-Fe-Ni-Co Alloy Coatings

Figure 6 and Figure 7 show the room temperature ^57^Fe and ^119^Sn CEMS spectra of the quaternary Sn-Fe-Ni-Co alloy coatings, respectively.

Both ^57^Fe and ^119^Sn Mössbauer spectra were decomposed into a broad sextet (indicated by red line in Figure 6 and Figure 7) and a doublet (indicated by blue line in Figure 6 and Figure 7), in accordance with previous research related to electrochemically deposited ternary and quaternary alloys [4,5,6,7,8,9]. The broad sextet components are dominant in each spectrum and are typical of ferromagnetic amorphous alloys (see also magnetization results in Figures 10–13). In the case of ^57^Fe Mössbauer spectra, the sextets can be considered a superposition of a large number of magnetic sub-spectra belonging to iron atoms being in slightly different micro-environments of alloying elements within the amorphous matrix. The minor paramagnetic components are considered to be amorphous paramagnetic phases at room temperature. The sextets in the ^119^Sn spectra also reflect ferromagnetic amorphous alloys via the transferred hyperfine fields, detected at tin nuclei surrounded by the magnetic transitional metal atoms (Fe, Co and Ni) within the amorphous matrix. The doublets in ^119^Sn spectra are associated with amorphous alloy phases being in the paramagnetic state at room temperature. The average Mössbauer parameters of the sextets and doublets can be seen in Table 3 and Table 4, and their variation with the deposition current density values and the composition can be seen in and Figure 8 and Figure 9.

The model fit of the relatively noisy CEM magnetic spectra inspired the applied model, whereas in the model used in [8,9], it was supplemented to allow the estimation of the average direction of the magnetic moment, too. The results obtained using this model are compatible with the magnetic results.

Figure 8 and Figure 9 show that both iron and tin hyperfine magnetic fields, *B*_Fe_ and *B*_Sn_, increase with the decreasing absolute value of current density. This is observed for both the DC and PP deposits. However, the average hyperfine magnetic field values were lower for the PP deposits than for the DC deposits. At the same time, the hyperfine fields show correlation with the change in the iron and tin content, which increases and decreases, respectively, as a function of the current density.

In the case of the ferromagnetic Fe alloys, the hyperfine magnetic field and its distribution can be simulated as a function of structure and composition, taking into account the effect of the alloying element [30,31]. Both the mean hyperfine fields and hyperfine field distributions obtained from the Mössbauer spectra of the DC deposited samples are in good agreement with those calculated for the quaternary alloys where each alloy is randomly distributed [8,9]. Consequently, the DC deposits can be considered as amorphous alloys having short-range ordering similar to a quasi-disordered solid solution of the alloying elements. This is consistent with the results obtained for DC electrodeposited Sn-Co-Fe, Sn-Ni-Fe and Sn-Fe-Co-Ni amorphous alloy deposits [6,8,9,30,32].

However, the mean hyperfine fields and hyperfine field distributions derived from the CEMS spectra of the PP deposits, which differ from those of the DC deposits even in the similar composition range, are not consistent at all with the random distribution of alloying elements [9]. This reveals that their short-range ordering is significantly different from the DC deposits.

The occurrence of the ferromagnetic phase of the deposits was estimated by determining the relative spectral area of the sextets (A_sextet_). In Table 3 and Table 4, it is shown that the ferromagnetic phase is the dominant phase in all deposits. The variation in the relative area of the sextet in ^57^Fe spectra of the DC deposits shows a decreasing tendency with increasing absolute value of current density, as well as with increasing Sn content. This indicates that higher tin concentration results in more paramagnetic phases in the amorphous deposits. The dependence of the relative areas of the sextet of the ^57^Fe spectra of the PP deposits slightly deviates from those of the DC deposits. This might be generated as a result of the different short-range orders forming during the DC and PP electrodeposition. The relative areas of the sextet in ^119^Sn spectra of PP show smaller values as compared to those of DC ^119^Sn spectra. By neglecting the difference between the f-factors in the ferromagnetic and paramagnetic phases, the difference may be connected to the fact that the ^119^Sn spectra reflect an additional tin-containing paramagnetic phase within the deposits, not observable in the ^57^Fe spectra.

The relative area of the second and fifth lines (A_2,5_) relative to the first and sixth lines (A_1,6_) of the Fe sextet provides information about the direction of the internal magnetic field, and thus about the magnetic anisotropy [33]. The angle *θ* between the γ-ray direction and the average direction of the magnetic moment of iron can be determined from the relative areas of the peaks A_2,5_ and A_1,6_ by:A_2,5_/A_1,6_ = 4sin^2^*θ*/(3 + 3cos2*θ*)

Although it is not so easy to identify the change in the relative area of the second and fifth lines of the sextet by comparing the envelopes of the ^57^Fe Mössbauer spectra of the deposits (Figure 6), by the fit used in the evaluation, it was possible to determine that the relative line areas and the *θ* angles characterizing the direction of magnetization calculated from them are different for deposits prepared at different current densities, which is shown as *M* in Table 3. It can be seen from Table 3 and Figure 6 that the average direction of magnetization varies between 66 and 73 degrees relative to the normal direction of the sample deposit, where angle *θ* increases apparently with increasing absolute value of current density for both the DC and PP samples, since the large errors in determination of angle *θ* does not allow these dependencies to be significant. However, the angles *θ* clearly demonstrate that magnetic anisotropy is present in all the electrodeposited quaternary Sn-Fe-Ni-Co alloy coatings. This can be associated with the shape anisotropy occurring in thin ferromagnetic metallic layers. Significant changes in the magnetic anisotropy with deposition current density were found for Sn-Fe, Fe-Co, Sn-Co-Fe and Sn-Ni-Fe electrodeposited alloys [1,2,6].

### 3.4. Magnetic Properties of the Quaternary Sn-Fe-Ni-Co Alloy Coatings

All six deposits measured showed similar magnetic features, and below, we present some typical data which generally reflect them all. For all deposits, isothermal magnetic moment *m*(*H*) plots were measured parallel and perpendicular to the surface of the deposits. For simplicity, all of the *m*(*H*)/*thickness* moments shown in Figure 10, Figure 11, Figure 12 and Figure 13 are given as emu/μm units, which can take into consideration the variation in the thickness of the deposits, presented in Table 2. *m*(*H*) values of the Cu substrates were measured at 5 K, and it was found that the substrates had negative magnetic moment values of 3–4 orders of magnitude lower than that of the deposits and negligible hereafter.

Figure 10 and Figure 11 show two typical experimental *m*(*H*) curves recorded for the DC1 and the PP3 deposited quaternary Sn-Fe-Ni-Co alloys, and their parameters are shown in Table 2. All the curves exhibit a ferromagnetic behavior with a pronounced anisotropy. The parallel *m*(*H*) curves increase sharply and are almost saturated around 2000 Oe, whereas the perpendicular *m*(*H*) branches saturate around 12,000 Oe, clearly indicating that (a) the parallel direction is the easy axis of the magnetization, which confirms the well-known evidence that in multi-layered magnetic deposits, the magnetic moments align in the coated planes; and (b) the difference in the saturation moment (*m_s_*) values between data measured at 5 and 295 K is ~17% for both directions, indicating that the magnetic transition temperature (*T_c_*) is well above room temperature; (c) at saturation, the *m_s_* for the perpendicular direction (the hard axis) is higher (by 10%) than *m*_s_ for the parallel one, whereas at lower fields, the parallel moments are higher, as presented hereafter (Figure 14). Similar behavior was obtained for the PP3 material (Figure 11), but here, the difference between *m_s_* measured in both directions decreases to 4% only. Higher *m_s_* along the hard axis was also observed in other Fe-Sn alloys and can be attributed to a slightly better aligned microstructure of the deposits [24].

As indicated earlier, all the alloy deposits exhibit almost the same magnetic nature. Figure 12 shows the *m*(*H*) plots measured at 5 K for all six deposits in the parallel direction, and for comparison, Figure 13 shows the same in the perpendicular direction measured at 295 K. Here again, the easy axis is the parallel direction. It is noticeable that in both cases, *m_s_* of the three DC deposits are higher than those of the PP quaternary Sn-Fe-Ni-Co alloy coatings. It is well accepted that the magnetic *m_s_* is directly proportional to the *B* values deduced from Mössbauer spectroscopy. Thus, the magnetic study confirms the higher *B* values listed in Table 3 and Table 4 for the DC deposits compared to those of the PP deposits.

All samples are soft magnets and their measured coercivity values are around 30–40 Oe. Due the different unavoidable remnant fields of the SQUID magnetometer, the uncertainties of the coercive fields are around 10–20 Oe.

The anisotropy exhibited here and explained below is typical for many alloys with magnetic anisotropy. Similar trends were found in oriented Fe_3_Sn and Fe_3_Sn_2_ alloys [21], in magnetically cooled Fe–49%Sn alloys [24] and also in thin epitaxial Co films [34].

Figure 14 and Figure 15 complete the information described above. (i) The magnetic susceptibility (*dm*/*dH*) of the PP1 deposit (Figure 14) measured in the parallel direction at 500 Oe is always higher than that measured in the perpendicular direction. (ii) The small changes in *m*(*T*) illustrate that *T_c_* of this material is well above room temperature. Moreover, Figure 15 shows a broad peak in the perpendicular direction, which is not observed in the parallel plot. The reason for this behavior is not yet known. It may reflect (i) a slight reorientation of the moments which are not completely aligned (below saturation), or (ii) the magnetic transition of some small particles which exhibit the paramagnetic doublets in the CEMS spectra (Figure 6 and Figure 7) or (iii) some structural changes.

In Figure 16 and Figure 17, the saturation magnetization is given by emu/g, where the mass of deposits *m* is shown in Table 2.

The dependence of the saturation magnetization of the DC-plated alloys (Figure 16) on the deposition current density shows a monotonically increasing dependence in both 5 K and 295 K measurements in perpendicular and parallel fields. This dependence shows a similar trend with the occurrence of the ferromagnetic phase, determined from the ^57^Fe and ^119^Sn CEMS measurements. This correlation, especially with those obtained from the ^57^Fe spectra, suggests that the saturation magnetization largely reflects the occurrence of the ferromagnetic phase. This is not surprising because in some steels with ferromagnetic and paramagnetic phases, the saturation magnetization is also used to measure the occurrence of the ferromagnetic phase [35]. As shown in Figure 16, the magnetization changes directly with the iron content, while also changing inversely with the tin content. This can be explained by the fact that as the absolute value of the current density increases, deposits with a higher tin content can be produced, in which a paramagnetic phase is formed to a greater extent. The existence of magnetic anisotropy in the DC deposits is clearly reflected in the differences between the saturation magnetizations obtained for fields applied perpendicularly and in parallel. A correlation can be observed at the same time between the saturation magnetization and the average direction of the magnetization, characterizing the shape anisotropy, derived from the ^57^Fe CEMS measurements, which may suggest that this may also play a role in the value of the saturation magnetization. A correlation of the saturation magnetization with the thickness of the deposit (Table 2) can also be possible, but in the case of the PP deposits (Figure 17), it seems to be more complex.

The dependence of the saturation magnetization on deposition current density of the PP deposited alloys (Figure 17) differs significantly from the dependence of the DC deposits shown in Figure 16. The saturation magnetization first increases and then shows a decreasing dependence on the current density for both 5 K and 295 K measurements both in perpendicular and in parallel field. This dependence shows a similar trend with the ferromagnetic phase content derived from the ^57^Fe CEMS measurements, thus confirming that the saturation magnetization mainly reflects the occurrence of the ferromagnetic phase. However, the change in saturation magnetization does not correlate with the change in ferromagnetic content determined by ^119^Sn CEMS, suggesting that the precipitation of the tin-containing phases and thus the formation of a short-range order are different from that of the DC deposits. This is also supported by the fact that the iron content and the saturation magnetization change similarly, while the tin content thus depends inversely on the current density. Furthermore, the current density dependence of the anisotropy obtained by the Mössbauer measurement is not similar to the current density dependence of the saturation magnetization, suggesting that the anisotropy detected by the magnetic measurements may not be influenced by the change of a few degrees in the average direction of magnetization in the studied range, which again may be due to differences in the structure of the PP and DC deposits.

The results of magnetization measurements are consistent with those obtained with electrochemical, SEM, EDX, XRD and Mössbauer spectroscopy measurements. Accordingly, quaternary Sn-Fe-Ni-Co alloys have been successfully synthetized with an amorphous structure either by DC and PP deposition, exhibiting (i) different short-range order and (ii) different morphology, as well as (iii) dominant ferromagnetic and minor paramagnetic composition. The magnetization measurements revealed a considerable magnetic anisotropy in all deposits, indicating low-temperature magnetic transitions in certain deposits, thus making the analysis of phase composition more accurate and contributing to the understanding of the differences between the DC and PP plated coatings.

The values of the specific magnetization are roughly in the range which was reported for Fe-Sn amorphous alloys [18]. According to the variations in the saturation magnetization on the different plating parameters (Figure 16 and Figure 17), the saturation magnetization mainly reflects the occurrence of a ferromagnetic phase in the deposits, but this is also influenced by several other parameters, such as the deposition conditions, the composition, the density, the short-range order, the thickness and the anisotropy. The lower saturation magnetization values of PP deposits than those of DC plated coatings may be due to the difference in density and thickness of the two types of coatings. The slight differences between the magnetization of PP samples may be explained with the structural and compositional changes in connection with changes as reflected in Figure 2.

The strong correlation between the magnetization curves and the ferromagnetic phase obtained from the ^57^Fe Mössbauer measurements mutually support and complement each other. Accordingly, the differences between the ferromagnetic phase determined by ^119^Sn Mössbauer spectroscopy compared to those determined by the ^57^Fe CEMS and magnetization measurement can indicate structural differences between the DC and PP deposits. Particularly, in the case of DC deposits, the occurrences of the ferromagnetic phase obtained with both methods are almost the same, except for the small decrease in the DC1 deposit measured with the ^119^Sn CEMS (Figure 16), and the short-range order of paramagnetic phase can be considered to be more or less similar to that suggested previously for the ferromagnetic amorphous alloy. However, the significant differences between the much lower content of the ferromagnetic phase in the PP deposits indicated by ^119^Sn CEMS compared to that obtained ^57^Fe CEMS and magnetization measurement (Figure 17) can also be related to the presence of segregated tin rich and iron poor amorphous paramagnetic phases.

The difference in magnetizations measured in magnetic fields applied parallel and perpendicular to the planes of the alloys (Figure 10, Figure 11, Figure 12 and Figure 13) clearly shows that magnetic anisotropy was formed in the deposits with amorphous structure. Magnetic anisotropy has already been found in a number of crystalline- and amorphous-related materials, and in electrochemically prepared materials [36,37,38,39,40,41,42,43,44]. The origin of magnetic anisotropy in amorphous phases was explained by a number of models [45,46,47]. In-plane uniaxial magnetic anisotropies arising in ferromagnetic amorphous alloy film deposits were considered to be consistent with bond-orientational anisotropy, related to microstructural mechanism reported in [47]. This may partly explain the magnetic anisotropy results and the differences in anisotropy between the DC and PP deposits being consistent with their different short-range orders and with the growth mechanism of the amorphous electrodeposits [48]. The magnetic anisotropy detected by the present magnetization measurements can also be related to the deposition current density, which is dependent on the average direction of the magnetic moment, determined by ^57^Fe CEMS, reflecting considerable anisotropy in both the DC and PP deposited quaternary Sn-Fe-Ni-Co alloys.

The ferromagnetic DC and PP electrodeposited quaternary Sn-Fe-Ni-Co alloy coatings are expected to be promising electrode material in Li-ion batteries.

## 4. Conclusions

The following conclusions can be drawn from the results of the magnetization, the ^57^Fe and ^119^Sn Mössbauer, the XRD, SEM and EDX measurements of the electrochemically prepared quaternary Sn-Fe-Ni-Co alloy coatings.

Quaternary Sn-Fe-Ni-Co alloy deposits can be successfully prepared by DC or PP electrochemical deposition from a dilute gluconate bath with amorphous structures in the composition range of 37–44 at% Sn, 35–39 at% Fe, 6–8 at% Ni and 13–17 at% Co, consisting of dominantly ferromagnetic phase alongside a minor paramagnetic phase.

Pronounced magnetic anisotropy was found in all the quaternary alloy deposits, reflected by the considerable differences in *m*(*H*) curves measured parallel or perpendicular to the applied fields, clearly indicating that the parallel direction is the easy axis of the magnetization.

The difference in saturation moment values between data measured at 5 and 295 K is ~17% (for DC1) for both directions, which indicates that the magnetic transition temperature (*T_c_*) is well above room temperature.

The *m*(*H*) plots in both parallel and perpendicular directions reflect that *m_s_* of the DC deposits are higher than those of the PP quaternary Sn-Fe-Ni-Co alloys, explaining the higher magnetic induction *B* values obtained from the Mössbauer measurements for the DC deposits compared to the PP alloy deposits.

The dependence of saturation magnetization of the electrodeposited quaternary Sn-Fe-Ni-Co alloy deposits on the deposition current density showed a correlation with the occurrence of the ferromagnetic phase, determined from the ^57^Fe and ^119^Sn CEMS measurements. The differences found between the curves of the DC and PP deposits can be associated with differences in the short-range ordering between the DC and the PP deposited quaternary Sn-Fe-Ni-Co alloys.

## Figures and Tables

**Figure 1 materials-15-03015-f001:**
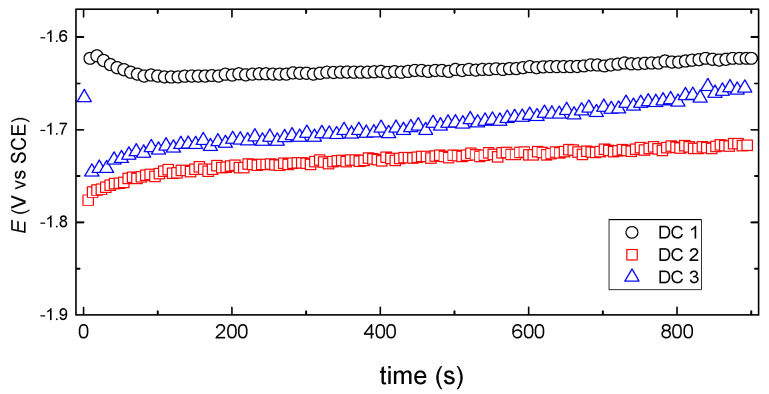
Cathode potential versus deposition time for DC deposited samples.

**Figure 2 materials-15-03015-f002:**
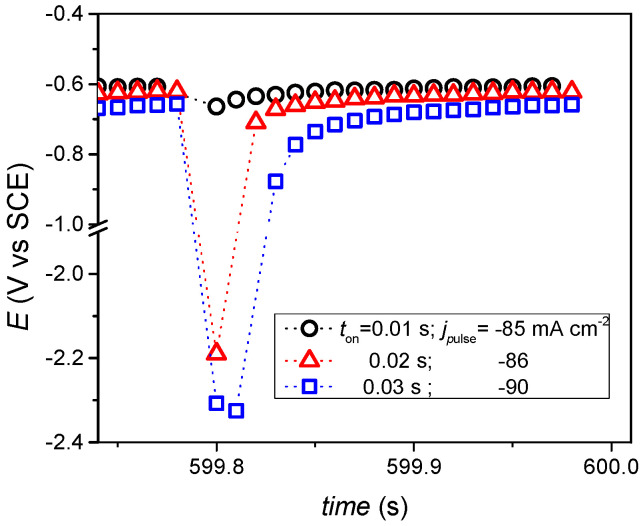
Typical cathode potential change (decay) during one period of pulse plating (PP).

**Figure 3 materials-15-03015-f003:**
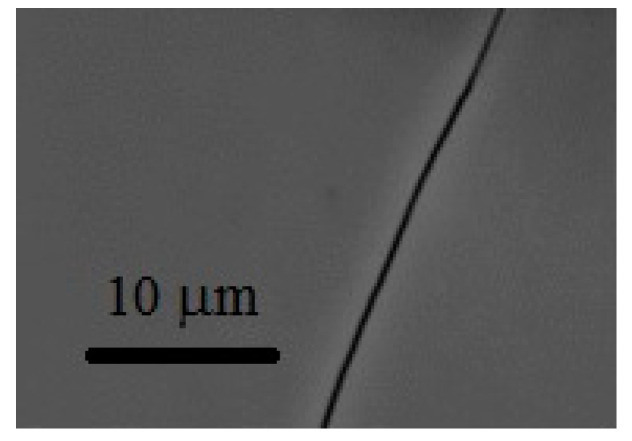
SEM micrographs (magnification 5000×) of DC3.

**Figure 4 materials-15-03015-f004:**
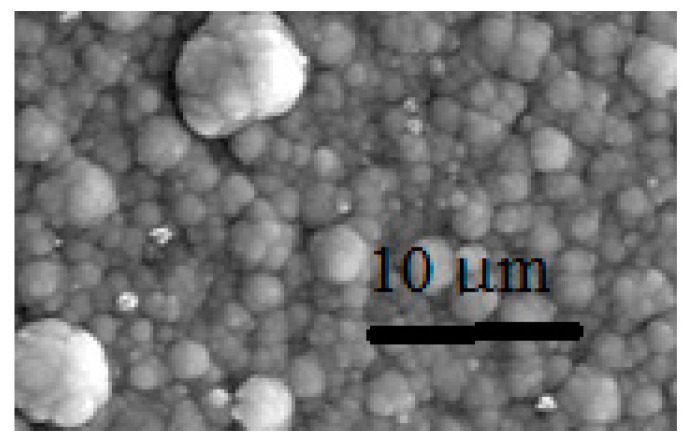
SEM micrographs (magnification 5000×) of PP3.

**Figure 5 materials-15-03015-f005:**
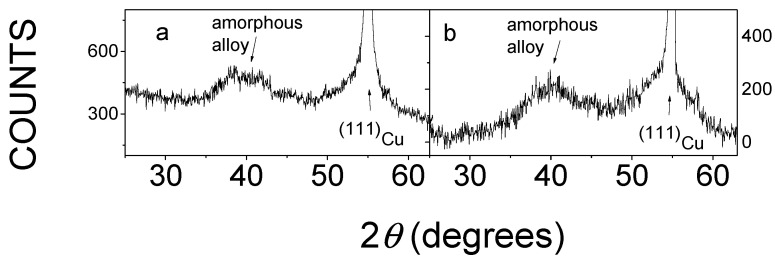
Details of XRD patterns of the DC3 (**a**) and PP3 (**b**) electrodeposited quaternary Sn-Fe-Ni-Co alloys.

**Figure 6 materials-15-03015-f006:**
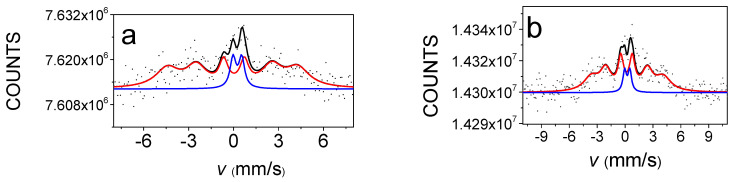
^57^Fe conversion electron Mössbauer spectra of DC3 (**a**) and PP3 (**b**) electrodeposited quaternary Sn-Fe-Ni-Co alloy coatings. The red and blue components show the ferromagnetic and paramagnetic amorphous phases, respectively, while the black line indicates their spectral superposition.

**Figure 7 materials-15-03015-f007:**
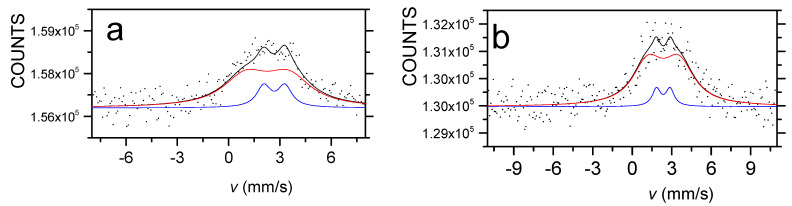
^119^Sn conversion electron Mössbauer spectra of DC3 (**a**) and PP3 (**b**) electrodeposited quaternary Sn-Fe-Ni-Co alloy coatings. The red and blue components show the ferromagnetic and paramagnetic amorphous phases, respectively, while the black line indicates their spectral superposition.

**Figure 8 materials-15-03015-f008:**
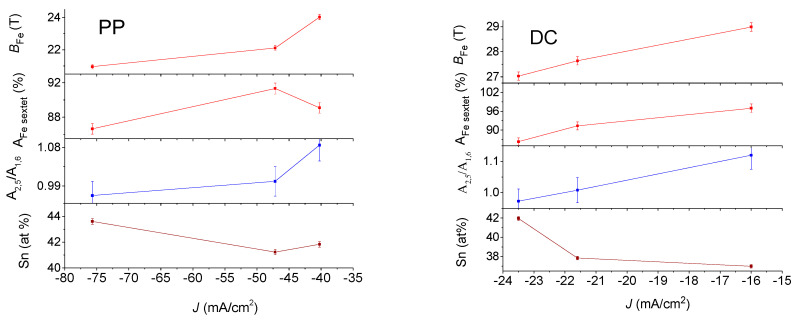
^57^Fe conversion electron Mössbauer DATA of DC (on the (**right**)) and PP (on the (**left**)) electrodeposited quaternary Sn-Fe-Ni-Co alloy coatings together with Sn content with the current densities.

**Figure 9 materials-15-03015-f009:**
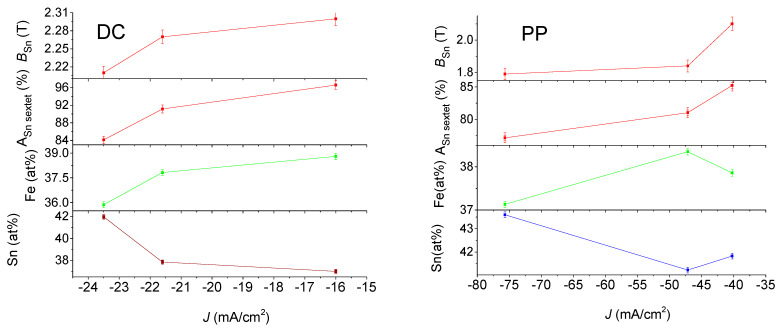
Variation in derived parameters from ^119^Sn conversion electron Mössbauer spectra of DC (on the (**left**)) and PP (on the (**right**)) electrodeposited quaternary Sn-Fe-Ni-Co alloy coatings together with Sn and Fe content with the current densities.

**Figure 10 materials-15-03015-f010:**
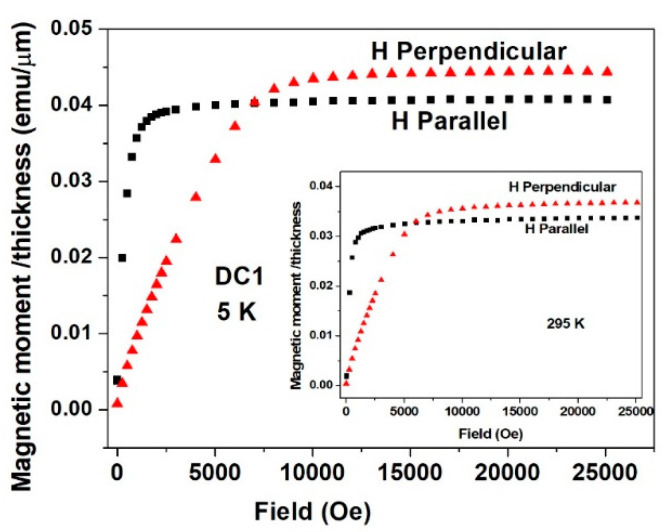
Isothermal magnetic moment plots measured at 5 and 295 K (inset) parallel (black symbols) and perpendicular (red symbols) to the plane of DC1 deposited quaternary Sn-Fe-Ni-Co alloy coatings. The moment values are in emu/μm (see text).

**Figure 11 materials-15-03015-f011:**
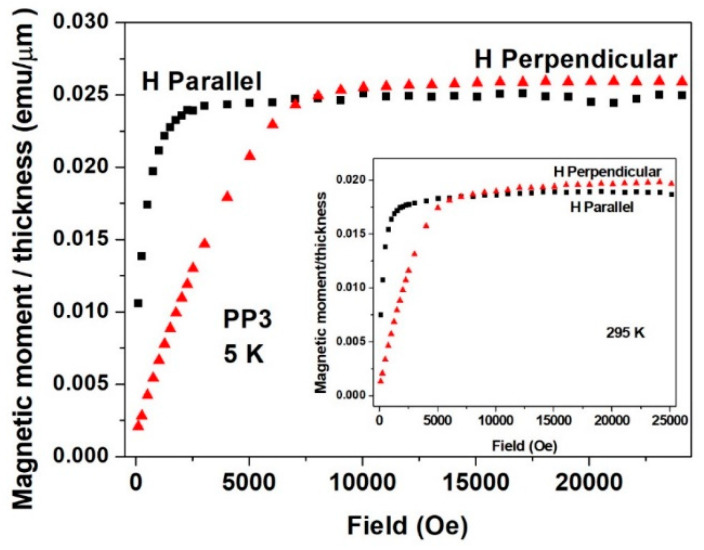
Isothermal magnetic moment plots measured at 5 K parallel (black symbols) and perpendicular (red symbols) to the plane of PP1 deposited quaternary Sn-Fe-Ni-Co alloy coatings. The inset shows the similar plots measured at 295 K.

**Figure 12 materials-15-03015-f012:**
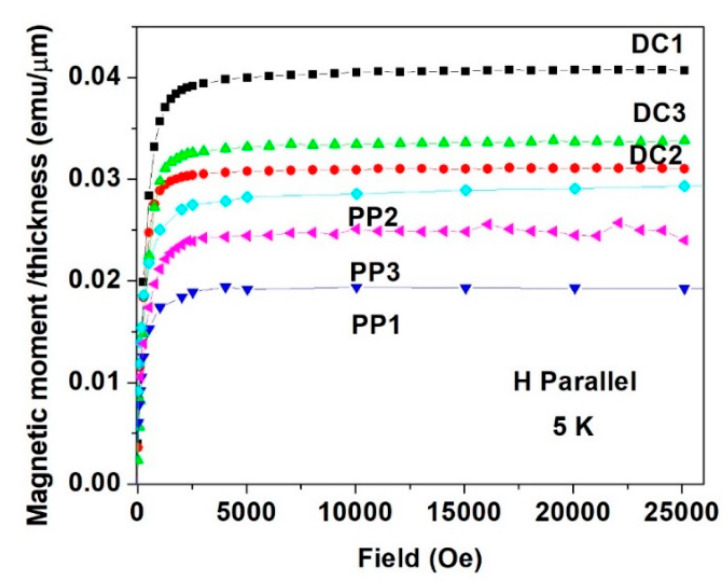
Parallel field dependence of the magnetic moment curves measured at 5 K for all DC and PP deposited quaternary Sn-Fe-Ni-Co alloy coatings.

**Figure 13 materials-15-03015-f013:**
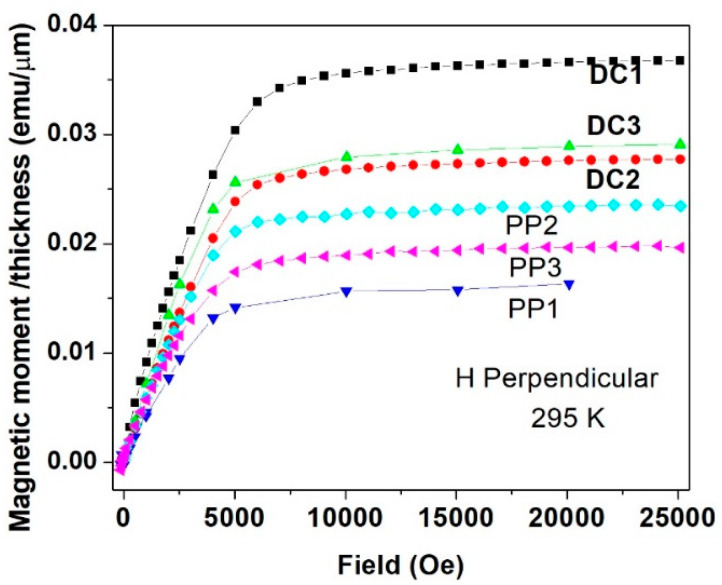
Perpendicular field dependence of the magnetic moments measured at 295 K for all DC and PP deposited quaternary Sn-Fe-Ni-Co alloy coatings.

**Figure 14 materials-15-03015-f014:**
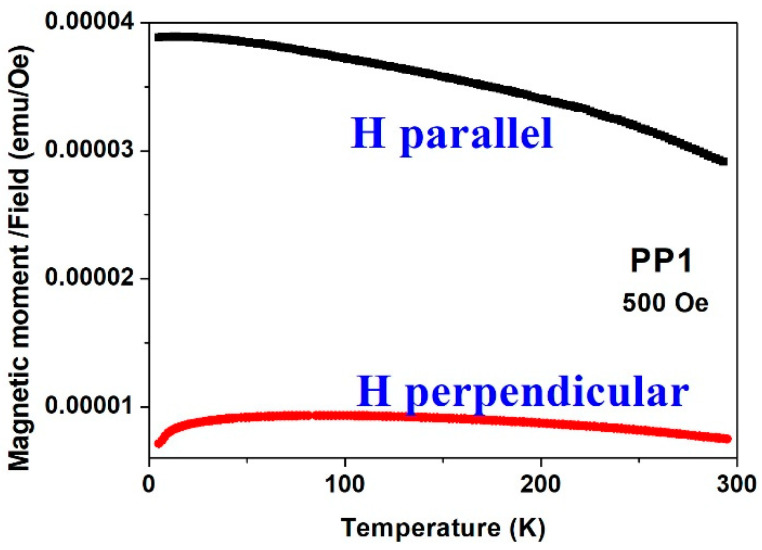
The temperature dependence of the susceptibility (*m*/*H*) under applied fields of 500 Oe measured parallel and perpendicular of PP1 sample.

**Figure 15 materials-15-03015-f015:**
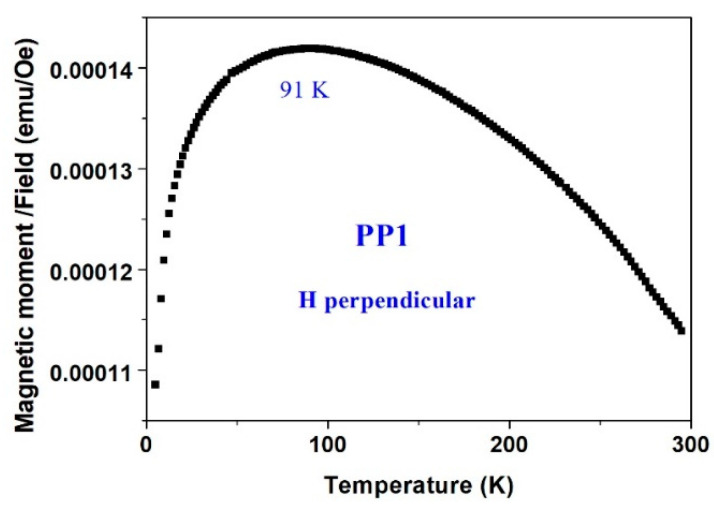
An extended scale of Figure 14 for the perpendicular susceptibility of PP1 sample.

**Figure 16 materials-15-03015-f016:**
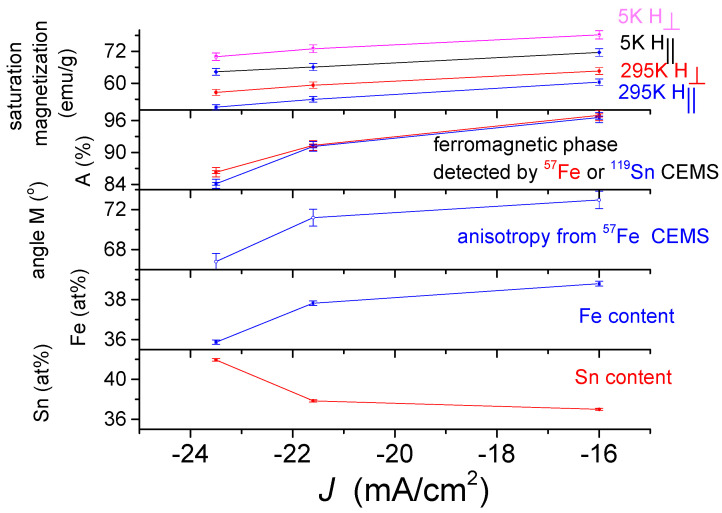
Dependence of saturation magnetization of DC electrodeposited quaternary Sn-Fe-Ni-Co alloy coatings on the current density, together with the current density dependences of the occurrence of ferromagnetic phase detected by ^57^Fe and ^119^Sn CEMS, direction of average magnetization originated from ^57^Fe CEMS, as well as Fe and Sn content.

**Figure 17 materials-15-03015-f017:**
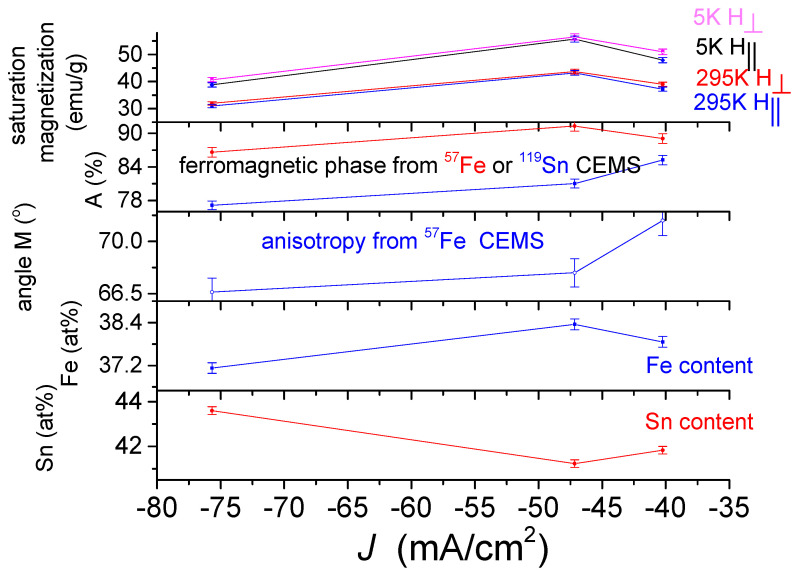
Dependence of saturation magnetization of PP electrodeposited quaternary Sn-Fe-Ni-Co alloy coatings on the current density, together with the current density dependences of the occurrence of ferromagnetic phase detected by ^57^Fe and ^119^Sn CEMS, direction of average magnetization derived from ^57^Fe CEMS, as well as Fe and Sn content.

**Table 1 materials-15-03015-t001:** Composition of the electrolyte and concentrations of the constituting compounds.

SnSO_4_ mol/dm^3^	FeSO_4_ mol/dm^3^	CoSO_4_ mol/dm^3^	NiSO_4_ mol/dm^3^	Sodium Gluconate mol/dm^3^	NaCl mol/dm^3^	H_3_BO_3_ mol/dm^3^	Ascorbic Acid mol/dm^3^	Peptone g/dm^3^
0.03	0.02	0.01	0.05	0.3	0.3	0.45	0.01	0.1

**Table 2 materials-15-03015-t002:** Selected deposition parameters and composition of DC and pulse current (PP) plated Sn-Fe-Ni-Co quaternary alloys *.

Sample/ Parameter	DC1	DC2	DC3	PP1	PP2	PP3
***J*****_dep_** (mA/cm^2^) (±1 μA/cm^2^)	−16.0	−21.6	−23.5			
***E*****_av_** (V vs. SCE) (±1 mV)	−1.63	−1.73	−1.70			
***J*_pulse ΣM_** (mA/cm^2^) **				−75.68	−47.16	−40.24
***E*_on_** (V vs. SCE) (±1 mV)				−1.76	−2.19	−2.33
Current efficiency H (±0.001)	0.396	0.395	0.409	0.890	0.547	0.446
***t*_on_** (s)				0.01	0.02	0.03
***t*_off_** (s)				0.19	0.18	0.17
Sn (at%)	37.0	37.8	42.0	43.6	41.2	41.8
Fe (at%)	38.8	37.8	35.9	37.1	38.4	37.9
Ni (at%)	7.4	8.0	7.6	6.0	6.7	6.6
Co (at%)	16.8	16.4	14.6	13.3	13.7	13.7
*d* (μm) (±0.001)	2.99	4.05	4.73	1.26	1.54	1.98
*m* (mg) (±0.001)	1.71	1.89	2.47	0.638	0.823	1.001

* The notations are the following: deposition current density, *J*_dep_; average potential, *E*_av_; pulse current density, *J*_pulse_; pulse-on potential, *E*_on_; pulse-on time, *t*_on_; pulse-off time, *t*_off_; thickness of deposit, *d*; mass of deposit, *m.* ** *J*_pulse_
_ΣM_ was calculated using current efficiencies and pulse-on current densities.

**Table 3 materials-15-03015-t003:** Mössbauer parameters derived from ^57^Fe CEMS of DC and PP electrodeposited quaternary Sn-Fe-Ni-Co alloys.

^57^Fe	*δ*_sextet_ (mm/s)	*B* (T)	A_sextet_ (%)	*δ*_doublet_ (mm/s)	*Δ*_doublet_ (mm/s)	A_doublet_ (%)	A_2,5_/A_1,6_	Angle of *M* (°)
DC1	0.13	28.99	96.98	0.25	0.57	3.02	1.12	72.9
	±0.02	±0.21	±1.34	±0.02	±0.04	±0.60	±0.02	±0.63
DC2	0.13	27.6	91.34	0.25	0.62	8.66	1.01	71.2
	±0.02	±0.19	±1.22	±0.02	±0.04	±0.78	±0.02	±0.61
DC3	0.13	27.03	86.3	0.24	0.58	13.7	0.97	66.8
	±0.02	±0.20	±0.96	±0.02	±0.04	±0.95	±0.02	±0.65
PP1	0.15	20.99	86.64	0.29	0.58	13.36	0.96	66.6
	±0.03	±0.30	±0.99	±0.04	±0.07	±0.89	±0.02	±0.60
PP2	0.13	22.12	91.32	0.24	0.55	8.68	1	67.9
	±0.03	±0.31	±1.17	±0.04	±0.08	±0.87	±0.02	±0.67
PP3	0.19	24.02	89.08	0.24	0.55	10.92	1.08	71.4
	±0.03	±0.29	±1.31	±0.04	±0.07	±0.71	±0.02	±0.62

A: relative spectral area, *δ*: isomer shift, *Δ*: quadrupole splitting, B: mean hyperfine magnetic field, A_2,5_/A_1,6_: relative areas of the 2nd and the 5th lines of sextets (A_2,5_), compared to those of the 1st and the 6th lines (A_1,6_), angle of *M*: angle between the directions of magnetic moment and γ-rays.

**Table 4 materials-15-03015-t004:** Mössbauer parameters derived from ^119^Sn CEMS of DC and PP electrodeposited quaternary Sn-Fe-Ni-Co alloys.

^119^Sn	*δ*_sextet_ (mm/s)	*B* (T)	A_sextet_ (%)	*δ*_doublet_ (mm/s)	*Δ*_doublet_ (mm/s)	A_doublet_ (%)
DC1	2.11 ± 0.03	2.30 ± 0.03	96.57 ± 0.76	2.28 ± 0.04	1.03 ± 0.07	3.42 ± 0.25
DC2	2.15 ± 0.03	2.27 ± 0.09	91.14 ± 0.72	2.31 ± 0.04	1.03 ± 0.07	8.85 ± 0.39
DC3	2.17 ± 0.04	2.21 ± 0.11	84.12 ± 0.74	2.35 ± 0.04	1.08 ± 0.07	15.87 ± 0.43
PP1	2.02 ± 0.04	1.79 ± 0.03	77.14 ± 0.85	2.31 ± 0.04	1.07 ± 0.07	22.86 ± 0.55
PP2	2.09 ± 0.04	1.84 ± 0.08	81.03 ± 0.88	2.3 ± 0.04	1.03 ± 0.07	18.97 ± 0.51
PP3	2.10 ± 0.03	2.10 ± 0.10	85.24 ± 0.83	2.3 ± 0.04	1.01 ± 0.07	14.75 ± 0.41

A: relative spectral area, *δ*: isomer shift, *Δ*: quadrupole splitting, *B*: mean hyperfine magnetic field.

## Data Availability

Relevant data have been shown in the paper.

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
