# Peer review of "Magnetic Anisotropy and Microstructure in Electrodeposited Quaternary Sn-Fe-Ni-Co Alloys with Amorphous Character"

_materials, 2022, doi:10.3390/ma15093015_

Round 1

Reviewer 1 Report

The present paper presents some interesting measurements about magnetic properties of Sn-Fe-Ni-Co alloy coatings. However, these alloys have been already studied by the same authors in previous publications (references [8] and [9]. Therefore, some of the conclusions of the paper are not novel. In special, the following ones:

a) "Quaternary Sn-Fe-Ni-Co alloy deposits can be successfully prepared". This is already known.

b) "The dependence of saturation magnetization of the electrodeposited quaternary Sn-Fe-Ni-Co alloy deposits on the deposition current density showed a correlation with the occurrence of the ferromagnetic phase,". This is a well known fact, so this correlation is not surprising.

From my point of view, the results, altough sound, are not interesting enough. They could be considered for publication in a different journal but not in Materials.

Author Response

ANSWER TO REVIEWER 1 (Materials 1619698)

The authors would like to thank the Reviewer 1  for his/her helpful comments.

We hope that  the following answers will satisfy him and lead to the publication of our article soon.

His/her comments were:

“ The present paper presents some interesting measurements about magnetic properties of Sn-Fe-Ni-Co alloy coatings. However, these alloys have been already studied by the same authors in previous publications (references [8] and [9]. Therefore, some of the conclusions of the paper are not novel. In special, the following ones: a) "Quaternary Sn-Fe-Ni-Co alloy deposits can be successfully prepared". This is already known. b) “The dependence of saturation magnetization of the electrodeposited quaternary Sn-Fe-Ni-Co alloy deposits on the deposition current density showed a correlation with the occurrence of the ferromagnetic phase,". This is a well known fact, so this correlation is not surprising.“

Our answer: Indeed, the novel Sn-Fe-Ni-Co alloy coatings were studied by us and reported in [8] and [9], when  the  deposits were successfully prepared for the first time. However, there  we did not discuss the occurrence of magnetic anisotropy deduced from the new dc magnetic studies, presented here. These new magnetic measurements, permit us  (i) to detect the magnetic anisotropy and its dependence on the production parameters, and (ii) to verify for the first time, the occurrence of the ferromagnetic coupling in these quaternary electrodeposits. The magnetic results  inspired us to use here a different model for the analysis  of the 57Fe Mössbauer spectra. Thus, the magnetic anisotropy deduced here of the Mössbauer spectra analysis is in full  agreement with the magnetic measurements.

This magnetic anisotropy of the Mössbauer spectra  and their dependence on the production parameters, also  highlighted the role of the presence of second phases The fact that the internal magnetic field measured by Mössbauer spectroscopy depends not only on the short-range order but also on the magnetic anisotropy is novel. Now we understand better  the difference between the structures of quaternary alloys prepared by DC and PP methods.

Reviewer 2 Report

The authors addressed all the suggested observations and improved the quality of the paper.

Author Response

ANSWER TO REVIEWER 2 (Materials 1619698)

The authors would like to express their thanks to the Reviewer 2 for the helpful comment.

His/her comment was:

“(x) English language and style are fine/minor spell check required“

The authors addressed all the suggested observations and improved the quality of the paper.

Our answer: We have checked the English and spelling, and we are ready to apply the service of the pulisher, if needed.

Reviewer 3 Report

The manuscript entitled “Magnetic anisotropy and microstructure in electrodeposited 3 quaternary Sn-Fe-Ni-Co alloys with amorphous character” described preparation and characteristic of electrodeposit amorphous material.

The manuscript is clearly written, the evaluation is properly described and relevant information is extracted. In my opinion it is important work suitable for publication in journal Materials. However, I don’t understand why only selected xrd patterns and Mossbauer spectra are presented in the manuscript. Therefore, before publication in Materials I suggest to add missing xrd pattern and Mossbauer spectra into the figures 5, 6 and 7. Simultaneously adding of marker of the Cu 111 and 200 reflections will be helpful for the readers for orientation in the figure.

After completing the data I believe the manuscript will be suitable for publication.

Author Response

ANSWER TO REVIEWER 3 (Materials 1619698)

The authors would like to express their thanks to the Reviewer 3 for helpful comments and suggestions improving the manuscript. We accepted all the Reviewer's remarks and corrected the manuscript accordingly.

His/her comments were:

“ The manuscript is clearly written, the evaluation is properly described and relevant information is extracted. In my opinion it is important work suitable for publication in journal Materials. However, I don’t understand why only selected xrd patterns and Mossbauer spectra are presented in the manuscript. Therefore, before publication in Materials I suggest to add missing xrd pattern and Mossbauer spectra into the figures 5, 6 and 7. Simultaneously adding of marker of the Cu 111 and 200 reflections will be helpful for the readers for orientation in the figure.

After completing the data I believe the manuscript will be suitable for publication.“

Our answer: We fully agree with and thankful for the suggestion of the Reviewer 2  to add missing XRD pattern and Mössbauer spectra into the figures 5, 6 and 7. We are ready to revise the manuscript accordingly. However we note that another Reviewer in the pre-peer-review process requested to change these figures which were included in the original manuscript.  Related to the other his suggestion, we added arrows to assign of the Cu (111) reflections in Figure 5. The (002) lines are out of the XRD pattern.

Reviewer 4 Report

The studies of the magnetic properties of amorphous alloys are welcome in the field of materials science. The present manuscript reports the study of the quaternary alloy Sn-Fe-Ni-Co synthesized by electrodeposition, extending and continuing the works published in https://doi.org/10.1007/s10751-017-1474-y and http://www.researchtrends.net/tia/abstract.asp?in=0&vn=21&tid=19&aid=6402.

To characterize the samples, such techniques as magnetometry, conversion electron Mössbauer spectroscopy, X-ray diffractometry, scanning electron microscopy and energy-dispersive X-ray spectroscopy were applied. In total, 6 samples with different synthesis conditions were investigated (3 for DC electroplating and 3 for pulsed plating). The variation of properties as a function of the current density is carefully discussed for both synthesis methods. The paper is coherent and well written, the analysis is exhaustive and convincing, the experimental results are carefully reported and the overall results are of interest to the community.

I recommend the manuscript for publication in Materials journal – below I just list some minor remarks to be addressed before publication:

* Regarding the beginning of the subsection 3.4. Magnetic properties of the quaternary Sn-Fe-Ni-Co alloy coatings – the magnetic moments in the plots are expressed as moment/thickness (emu/μm). I guess this selection could be discussed in a little bit more extensive manner. Typically, the moment is expressed in emu/g (or emu/cm^3) and this is volume or mass magnetization density. If the total measured moment is divided just by the thickness, it seems that it remains a quantity dependent on the surface area of the sample.  I believe some explanation would be valuable.

* Figures 6 and 7 might contain the explanation of different curves (black/red/blue) also in the figure label (to supplement the explanation given in the text).

Author Response

ANSWER TO REVIEWER 4 (Materials 1619698)

The authors would like to express their thanks to the Reviewer 4 for helpful comments and suggestions improving the manuscript. We accepted all the Reviewer's remarks and corrected the manuscript accordingly.

Reviewer 4 comments were:

“To characterize the samples, such techniques as magnetometry, conversion electron Mössbauer spectroscopy, X-ray diffractometry, scanning electron microscopy and energy-dispersive X-ray spectroscopy were applied. In total, 6 samples with different synthesis conditions were investigated (3 for DC electroplating and 3 for pulsed plating). The variation of properties as a function of the current density is carefully discussed for both synthesis methods. The paper is coherent and well written, the analysis is exhaustive and convincing, the experimental results are carefully reported and the overall results are of interest to the community.

I recommend the manuscript for publication in Materials journal – below I just list some minor remarks to be addressed before publication:

* Regarding the beginning of the subsection 3.4. Magnetic properties of the quaternary Sn-Fe-Ni-Co alloy coatings – the magnetic moments in the plots are expressed as moment/thickness (emu/μm). I guess this selection could be discussed in a little bit more extensive manner. Typically, the moment is expressed in emu/g (or emu/cm^3) and this is volume or mass magnetization density. If the total measured moment is divided just by the thickness, it seems that it remains a quantity dependent on the surface area of the sample.  I believe some explanation would be valuable.

* Figures 6 and 7 might contain the explanation of different curves (black/red/blue) also in the figure label (to supplement the explanation given in the text). “

Our answer: Indeed, it is most common to give the magnetic moment in units of emu /g or emu/cm3, but it is not entirely unprecedented to give the depth dependence (emu/cm). In our case, too, we found the latter (magnetic moment/thickness) to be the most expressive and simple, primarily with respect to the change in the thickness of the samples (given in Table 2) and the illustrating of the magnetic field changes applied perpendicularly and parallel to the surface of the samples.

The text was modified as indicated by yellow highlight:

In Page 4:

Magnetization measurements of electrodeposits were performed at the Hebrew University, using a commercial SQUID magnetometer MPMS-5S (Quantum Design) in the temperature range between 5 and 300 K. The isothermal field dependent magnetic moment m(H) curves were measured at 5 and 295 K with applied magnetic fields (H) up to 32 kOe. During the measurements, the surface of the deposits was oriented parallel and perpendicular to the applied external magnetic field, H. The m(H) values were normalized to the electrodeposits areas and exhibited as emu/d, where d is the thickness in (mm) as listed in Table 2.

in 3.4. in Page 13:

For simplicity all of the m(H)/thickness moments shown in Figures 10-13 are given as emu/mm units, which can take into consideration  the variation in the thickness of the deposits presented in Table 2.

in Figures 6 and 7, the explanation of different curves was included  in the caption of the figures:

The red and blue components show the ferromagnetic and paramagnetic amorphous phases, respectively, while black line indicates their spectral superposition.

Reviewer 5 Report

The authors present results of magnetic and structural characterization studies on quaternary Sn-Fe-Co-Ni coatings by both direct current and pulse plated deposition on copper substrates.  Results show significant differences in both structure and behavior between DC and pulse plated coatings.  However, both types of coatings show magnetic anisotropy.  Overall, the papers are well-written and understandable. 

One minor change is in the labeling of Table 2.  There are several statements and sentence fragments below the table.  It is not clear from the formatting if these were intended to be footnotes to the table, part of the general narrative or something else.  They should be reformatted and presented more clearly (e.g., the method used for Table 3).

Also, in Figures 6 and 7, include the explanation for the different lines in the caption of the figures themselves, not just in the narrative.

Author Response

ANSWER TO REVIEWER 5 (Materials 1619698)

The authors would like to express their thanks to the Reviewer 5 for helpful comments and suggestions improving the manuscript. We accepted all the Reviewer's remarks and corrected the manuscript accordingly.

Reviewer 5 comments were:

“ The authors present results of magnetic and structural characterization studies on quaternary Sn-Fe-Co-Ni coatings by both direct current and pulse plated deposition on copper substrates.  Results show significant differences in both structure and behavior between DC and pulse plated coatings.  However, both types of coatings show magnetic anisotropy.  Overall, the papers are well-written and understandable.

One minor change is in the labeling of Table 2.  There are several statements and sentence fragments below the table.  It is not clear from the formatting if these were intended to be footnotes to the table, part of the general narrative or something else.  They should be reformatted and presented more clearly (e.g., the method used for Table 3). Also, in Figures 6 and 7, include the explanation for the different lines in the caption of the figures themselves, not just in the narrative.“

Our answer: We amended several statements in Table 2, accordingly. We put the following text into the footnote of Table 2:

*The notations are the following: deposition current density, Jdep, average potential, Eav, pulse current density, Jpulse, pulse-on potential, Eon, as well as pulse-on time, ton, and pulse-off time, toff, thickness of deposit, d, mass of deposit, m. **Jpulse SM was calculated using current efficiencies and pulse on current densities.”

We left space enough to separate Table 2 from the narrative part.

in Figures 6 and 7, the explanation for the different lines was included  in the caption of the figures:

The red and blue components show the ferromagnetic and paramagnetic amorphous phases, respectively, while black line indicates their spectral superposition.

Reviewer 6 Report

This manuscript reports the growth of Sn-Fe-Ni-Co quaternary alloys using the DC and pulse plating (PP) electrodepositions and their structural and magnetic characterizations. The authors find that the structural differences between the DC and PP deposits are correlated to the magnetic differences. Besides, the compositional and magnetic properties as a function of current density have been well characterized. Overall, the manuscript contains genuine scientific results, which could be published in materials without delay.

I just have a small concern. Moessbauer fittings look bad. Some comments on the model reliability are needed.  

Author Response

ANSWER TO REVIEWER 6 (Materials 1619698)

The authors would like to express their thanks to the Reviewer 6 for helpful comments and suggestions improving the manuscript.

We accepted all the Reviewer's remarks and corrected the manuscript accordingly.

Reviewer 6 comments were:

“ This manuscript reports the growth of Sn-Fe-Ni-Co quaternary alloys using the DC and pulse plating (PP) electrodepositions and their structural and magnetic characterizations. The authors find that the structural differences between the DC and PP deposits are correlated to the magnetic differences. Besides, the compositional and magnetic properties as a function of current density have been well characterized. Overall, the manuscript contains genuine scientific results, which could be published in materials without delay.

I just have a small concern. Moessbauer fittings look bad. Some comments on the model reliability are needed.  “

Our answer: The Reviewer 6 is right.  We added the requested comment after Table 3 in Page 9:

The model fit of the relatively noisy CEM magnetic spectra inspired the applied model, whereas in  the model used in [8,9] it was supplemented to allow the estimation of the average direction of the magnetic moment, too. The results obtained using this model is compatible with the magnetic results.

Round 2

Reviewer 1 Report

As I said in my previous report, although the scientific results are good I am from the opinion that they are marginal and do not represent a significant advance in knowledge. Thus, I am from the opnion that these results can be published but not in a journal of the first quartil like Materials.

This manuscript is a resubmission of an earlier submission. The following is a list of the peer review reports and author responses from that submission.

Round 1

Reviewer 1 Report

In the paper it is presented an elaborate study on Sn-Fe-Ni-Co quaternary alloys prepared by two electrodeposition methods: direct current DC and pulse plating. The studies are based on materials characterization techniques and also, on magnetic measurements. The paper is interesting, but there are some corrections to be made in order to improve the overall quality.

Some observations regarding the English language corrections are provided in the attached file.

Please use the same way to describe the measuring units for different quantities in tables and figures. I recommend using in all figures and tables the format with round parentheses "(measuring unit)".

In all the paper, figures, and tables use the same style to represent the measuring units with fractional terms. Use either "XY-z" or "X/Y". 

Refer in the text the figures with "Figure. X ..." according to the template.

Some technical and scientific observations:

  1. Even some abbreviations are well known you must describe them as they first appear in the text.
  2. EDAX is the name of the company, not the abbreviation of the method. Use EDS or EDX.
  3. The "Experimental procedure" section must be completely rewritten. It is very confusing for the reader. You must explain which material was used for the based samples. What is the geometry on this samples? Also, explain the platting method. There are four layers that have to be deposited? In which order? 
  4. Use for current density "J" in tables and figures.
  5. Describe clearly in 3.4. section how are the samples placed in the applied external magnetic field. How is this related to the anisotropy of the samples?
  6. Please use in the text and figures the term "magnetic moment", not "moment". If the figures present magnetic moment values so use in the text "m" as abbreviation. Do not refer to magnetization as in M(H) and present values for magnetic moment. To conclude, reconsider in the text to use "m" as for magnetic moment instead of "M" for magnetization, or change all vertical axes and the values from figures 10 to 13 to consider the magnetization values.
  7. The vertical axes of figures 10, 11, 12, 13 have the measuring unit written wrong. It is not "emu/d". It must be "emu/m" because d is the abbreviation letter for the quantity thickness, but the measuring unit for thickness is "meter". The correct description of the axes is "magnetic moment/thickness (emu/m)"
  8. In figure 12 use line+symbol style as in figure 13 
  9. In figure 10 point out, on the figure or in the caption, the 5K and 295K curves. Maybe you can insert a legend in the figure.
  10. In Figures 8 and 9 the axes text is too small. Maybe an increase scale for the whole picture.
  11. For figures 16 and 17 increase the font size and the width for the vertical axis  

Reviewer 2 Report

Dear authors,

I really enjoyed reading your interesting manuscript on quaternary amorphous thin films. Here are some comments on your publication:

"SEM-EDAX": EDAX also works, as I have learned from Wikipedia right now. EDX, however, is much more common. Furthermore EDAX might mislead towards the company of EDAX: https://www.edax.com/ Please correct all EDAX to EDX.

Line 63: "magnetic" Please check, but I think that "magnetic" can be removed.

The two materials YbMnFeSn and CrMnFeCoNi are far off from the materials you used. The short section on these materials can be omitted.

In many cases, tables and figures extend to the next page. Try to fit tables and figures on one page.

In chapter 2 you write "...and the thickness of the deposits was calculated using the plating pa-118 rameters..." Please comment on the given accuracy (0.001µm according to table 2) of this approach as it seriously influences the calculation of emu/d and the comparison of different deposits. e.g. in Fig12 and Fig 13

Please provide, which EDX-device was used for measuring the chemical compositon.

Figure 1: As said above, the figure caption has to be on the same page as the figure.

Figure 3, Figure 4: Magnification does not tell anything. It depends on the ratio of scanned area and the size of display. The latter can differ very much. A valid micron bar has to be provided.

Figure 6, 7: Missing description / legend for black, red and blue lines.

Line 329. to be more precise: whereas beloe saturation of the perpendicular direction.

Is it possibel to give an explanation for the difference in saturation magnetization for two different measurement directions for M(H), but measured at the same temperature.

Figuer 11: Is it possible to draw Figure 11 alike Figure 10, without inset but using open and closed symbols. It will make comparison easier. 

Line 361: In case it is the same as in Figure 10, it might be that the magnetic moment in parallel direction is higher than in perp. direction. However, in Fig 14&15 you are showing results on susceptibility

Figure 16: Rework the graph. It is not a very nice on. E.g. numbers on y-axis are on top of each other. There is a couple of different font sizes, ... Adjust the font sizes and rearrange the y-axis to make it clear but also appealing.

Figure 17: Same comment as for Figure 16.

Some comments on the writing:

Line 50: "alloy DC deposited alloy" remove first alloy

Line 107: scovered -> covered

Table 2: Indices are missing, e.g. for "jdep"

Figure 14: text within the figure -> parallel

Line 369: Ann -> An

Reviewer 3 Report

The present paper basically studies the magnetic properties of a series of electrodeposited coatings. Besides this magnetic characterization, the authors make an attempt to correlate these properties with the structural information obtained from Conversion Electron Mössbauer Spectroscopy (CEMS). Although some of the conclusions are new and could be published, some other conclusions presented as novel are already shown in previous papers by the same authors. Moreover, from my point of view, the Mössbauer spectra are too noisy to be able to yield meaningful conclusions. In the following lines I will detail my concerns on the paper that I cannot recommend for publication in Materials.

A) In the Introduction the authors recognize that Mössbauer Spectroscopy is a useful technique to characterize the structure of electrodeposited coatings and they cite several of their previous papers. However, the final paragraph is misleading for the following reasons:

i) the quaternary alloys studied here are not “novel” as they were already presented in the references [8] and [9]. It should be recognized that in the present paper they also show different experimental procedures to obtain slightly different coatings, but the main composition is not new;

ii) it is said that the magnetic properties are studied as a function of the structure, composition and preparation parameters. From my point of view all these three variables are strictly correlated because the change in the preparation parameters leads to a different composition and the use of galvanostatic or pulse current (that are also preparation parameters) leads to different structures. Thus, what should be said is that the magnetic properties are studied as a function of the preparation parameters.

B) As commented in A), this paper does present different coatings prepared by galvanostatic or pulse current and in each case with different deposition parameters. These changes in the deposition parameters are novel of this paper but, from my point of view, the SEM and XRD results are equivalent to the already presented in [8] and [9], thus they do not provide any further insight in the characterization of the coatings. The authors can argue that the Mössbauer results do present differences that can be correlated with the preparation parameters. However, as the authors already recognize in the paper in line 297 (“it is not easy to identify the change in the relative area of the second and fifth lines of the sextet”), the quality of the spectra is not enough to show differences between the spectra of coatings obtained with different parameters. In this regard, I am of the opinion that the errors shown in Table 3 and 4 are too small. Thus, if it is not possible to improve the signal to noise ratio of the measured spectra I will not trust any of the results derived from the apparent differences in the spectra. This include all the discussion on the direction of magnetization (lines 289-309) and the correlation between the saturation magnetization and the CEMS results (lines 380-399).

C) Of all the conclusions presented in section 4, only the ones that make reference to the magnetic properties are conclusions of the present paper. The novelty of the alloy deposits (lines 478-481) is not true as these alloys have already been produced. The differences found between the curves of the DC and PP deposits (line 496-498) are also already presented in reference [8]. And, as explained before, the conclusion relating the saturation magnetization and the occurrence of the ferromagnetic phase determined by CEMS is disputable.

D) Section 3.4, regarding the magnetic properties, is a little bit disorganized. I am from the opinion that the notation should be unified (almost each figure present a different notation) and also the style (figures 10 and 11 present the same information for DC and PP coatings respectively but the information is shown differently). Moreover, the information contained in the figures is repeated and figures 10, 11, 12 and 13 present exactly the same data points. I would suggest to unify them. Finally, following my comments on B), figures 16 and 17 does not yield novel information as the errors bars should be larger.

Summarizing, I am from the opinion that all the sections of the paper that not deal with the magnetic properties are not new and should be removed. Thus, the parts that are really new I think that they are interesting enough to be published but I do not feel it matches the standards of the journal Materials and I would not recommend for publication in it.

Reviewer 4 Report

This paper is very decent experimental work on the novel electrodeposited quaternary Sn-Fe-Ni-Co alloys studied with a variety of techniques such as XRD, 57Fe and 119Sn conversion electron Mössbauer, SEM-EDAX and magnetization measurements. In general, it is a high-quality manuscript, but it can be accepted only after minor revision according to the remarques given below.

In the Abstract, Introduction and Conclusions the applicational information on the deposited alloys should be given. It is not clear what was the main goal of the work: to make alloys of unique (e.g. magnetic) features or to optimize (economize) the manufacturing process of well-known, but desired alloys.

The Authors should explain, why they did not try to measure coercivities. Presumably, the magnetic softness of the alloys (being a consequence of their amorphous structure) would be an applicational feature worthy underlining.

Authors should specify more precisely the origin and mechanism for the observed magnetic anisotropy.

Do the measurements of the magnetization curves at different temperatures as well as 57Fe and 119Sn Mössbauer spectra allow to estimate what is an Sn contribution in total magnetic moment?

In the Experimental procedure section, it should be specified in what way the hyperfine field distributions are considered in fitting procedures of the Mössbauer spectra.

Tables 3 and 4 should be supplemented with this information on the width of hyperfine field distributions.

In discussion and conclusions, it should be clearly stated if the applied electrodeposition process is really competitive to others technologies in the case of Sn-Fe-Ni-Co alloys.

English is good enough but requires minor corrections concerning the usage of the articles.

The list of references is complete and up-to-date.

Round 2

Reviewer 1 Report

The authors respond to most of the indicated suggestions from version 1 and improve the paper, but in the attached file I point out some additional corrections.

Reviewer 3 Report

I appreciate the answers provided by the authors to my queries and I consider that the paper has improved with respect the past version. However, I am from the opinion that my major concern (that the quality of the Mössbauer data is not enough to derive the conclusion of the paper) is still present. It is clear that the Mössbauer data is useful to derive the relative areas of the ferro and paramagnetic phases but: a) there is not enough statistics to obtain information on the intensities of the lines and b) with such level of noise the error bars should be larger, and this affects the conclusions. Of course I believe the authors when they say that they have checked the errors and are correct, but one thing is the error obtained from the computer fitting software (a pure mathematical error) and a different thing is the error with phyiscal meaning.

In conclusion, I still think that the paper is not suitable for Materials.
